# Oral-Health-Related Quality of Life and Cosleeping: The Role of Nocturnal Breastfeeding

**DOI:** 10.3390/children8110969

**Published:** 2021-10-26

**Authors:** María Carrillo-Díaz, Laura Lacomba-Trejo, María Pérez-Chicharro, Martín Romero-Maroto, María José González Olmo

**Affiliations:** 1Department of Nursing and Dentistry, Rey Juan Carlos University, 28922 Alcorcón, Spain; maria.carrillo@urjc.es; 2Department of Personality, Assessment and Psychological Treatments, Faculty of Psychology and Speech Therapy, Universitat de València, Av. Blasco Ibáñez, 21, 46010 Valencia, Spain; laura.lacomba@uv.es; 3Department of Orthodontics, Rey Juan Carlos University, 28922 Alcorcón, Spain; maria.perez@urjc.es (M.P.-C.); martin.romero@urjc.es (M.R.-M.)

**Keywords:** breastfeeding, cosleeping, oral-health-related quality of life, early childhood caries, oral health, child relations

## Abstract

The purpose of this paper is to analyse the association between cosleeping and the number of breastfeeding sessions in infants, OHRQoL of the child and the family, and the DMFT Child’s index. The sample comprised 273 children (2–4 years old). In addition to the clinical examination of the child to assess the DMFT Index, the mother was requested to complete a questionnaire to collect data about the breastfeeding practice, diet, dental hygiene, dental check-ups, quality of the child’s oral life, and family impact (ECOHIS Scale). The children’s OHRQoL is positively correlated with number of night-time breastfeeding sessions at 12 months (r^2^ = 0.40 **), DMFT index (r^2^ = 0.60 **), impact family (r^2^ = 0.65 **), and duration of cosleeping (r^2^ = 0.36 **). The moderating effect explained 41% of OHRQoL; the interaction between the number of breastfeeding sessions at 18 months and the DMFT index significantly increased the coefficient of determination. A longer practice time for cosleeping was associated with an increase in breastfeeding sessions, a higher impact on OHRQoL, a higher family impact, and a higher DMFT index. More than three night-time breastfeeding sessions moderate the relationship between the DMFT index and the child’s OHRQoL.

## 1. Introduction

Early childhood caries is the presence of one or more decayed (non-cavitated or cavitated lesions), missing (due to caries), or filled primary teeth in children aged 71 months (5 years) or younger. Early childhood caries is a chronic disease with a high prevalence that affects the oral-health-related quality of life (OHRQoL) and children’s parents [1]. OHRQoL is defined as the degree to which oral problems affect a patient’s psychosocial functioning and well-being [2]. In children, cavities can affect eating, sleeping, pronouncing some words, and smiling; it also might reduce school attendance; on the other hand, it also affects the family’s economy, as recurrent dental appointments are required, and absences at work cause guilt or other concerns for the family [3].

Some studies claim that human milk is less cariogenic than bovine/infant formulas [4], although there is controversy regarding this claim [5,6]. However, when breastfeeding is maintained beyond 12 months, especially at night, it is associated with an increased risk of dental caries development and even promotes the development of early childhood caries [7,8]. This is explained by the decreased salivary flow of the baby at night [8].

Infants who are breastfed at night in association with cosleeping wake-up are breastfed more times during the night. The time between feedings is shorter than that of other non-breastfed infants [9,10].

Cosleeping is defined as an infant and adult sleeping on the same sleep surface during the night-time or main sleep period. Cosleeping is a characteristic practice of mammals that provides many benefits and is recommended by the World Health Organization (WHO) and United Nations International Children’s Fund as part of the Baby Friendly Hospital’s Initiative program [11,12]. Cosleeping plays an important role in the promotion and duration of breastfeeding, in addition to other benefits such as controlling infant irritability or illness, improving the baby’s and parent’s sleep quality, feeding the emotional needs of the child through maternal contact, and allowing sleep to be established more quickly [13]. Cosleeping also results in long-term benefits such as improved social skills, esteem, or neuro-affective responses and reduces fears, tantrums, or even anxiety to stress in adult life [12].

One of the main reasons for supporting the implementation of cosleeping is that it facilitates breastfeeding. The World Health Organization (WHO) recommends immediate initiation of breastfeeding from the first hour of life, with exclusive breastfeeding up to 6 months of age and continued complementary breastfeeding thereafter up to 2 years. The WHO also advises that breastfeeding should be “on demand” (as many times as the child wants), day and night [14]. The benefits of breastfeeding for the overall health of the infant are clear, including reduced risk of mortality from infections; or sudden death; protection against gastrointestinal, respiratory, urinary, and otitis diseases; prevention of growth deficits in the first month of life and dental and skeletal malocclusions to facilitate good growth and craniofacial development; and many other psychological and immunological benefits. For these reasons, breastfeeding, together with cosleeping, is a common practice in all cultures; cosleeping facilitates night-time breastfeeding [14]. However, adverse effects of this practice have also been identified, such as an increased risk of sudden unexpected death in infancy (SUDI) or sudden infant death syndrome (SIDS) [15] when cosleeping is associated with unstable sleep, smoking, or parental drug use [16].

On the other hand, we consider that paediatric dentists play an important role in advising the child’s relatives about the dental benefits and risks derived from the practice of night-time breastfeeding. Dentists can provide parents with some strategies that can help to reduce their anxiety levels. In addition, parents can maintain cosleeping until their children acquire the necessary strategies to sleep on their own. This will, in turn, minimize the child’s oral health risk.

Considering the gaps in the literature, the general objective of this study is to analyse the association between cosleeping and the number of night-time breastfeeding sessions in infants (at 12 and 18 months of age), the impact on OHRQoL, and the Decay-Missing-Filled Teeth (DMFT) index of the child.

The questions we asked in our research are as follows: is there a significant relationship between cosleeping, the number of night-time breastfeeding sessions, and their OHRQoL? Is there a significant relationship between cosleeping and the child’s DMFT index? Finally, can the number of breastfeeding sessions at 18 months moderate the relationship between the child’s DMFT Index and his or her OHRQoL?

## 2. Materials and Methods

### 2.1. Sample

Our sample comprised 273 children (149 girls and 124 boys) residing in the southern area of the Autonomous Community of Madrid (Spain). Their average age was 3.01 years (standard deviation (SD) = 0.811), ranging from 2 to 4 years old.

The data were collected from regular individual visits of paediatric patients attending a Clinic in the months of November and December 2019. This clinic offers regular dental treatments such as check-ups, caries treatment, and orthodontics for children and adults. Participants in this study received a free oral health check-up.

The instructions for completion were provided by a member of the research team. Informed consent was obtained from all mothers involved in the study. Ethical approval for this study was obtained from the Ethics Commission for Research at the Rey Juan Carlos University (code 2409201913019).

Criteria: children between 2 and 4 years of age who agreed to participate and whose parents had signed the consent form.

Excluding criteria: Children who did not cooperate in the dental examination and those with systemic diseases or receiving pharmacological treatment were excluded.

### 2.2. Measures

In addition to the clinical examination of the child, a questionnaire was issued to the mother to be filled up while a research team member supervised the action to attend to any query that the mother might have. The aim of this questionnaire was to collect data on the practice of breastfeeding, type and duration of breastfeeding, diet, hygiene, dental check-ups, quality of the child’s oral life, and family impact. The main variables analysed in our study were measured as follows:

The clinical examination was carried out in the clinics’ dental chairs of the aforementioned centre. The examination materials consisted of sterile Community Periodontal Index (CPI) mirrors and probes, latex gloves, masks, and protective glasses. DMFT index values were recorded (decayed and filled temporary teeth).

A dichotomous question (yes/no) was used to ask parents if they had completed bedding with their child and the duration of the bed-sharing in months.

In relation to sugar consumption, a question was asked: “Does your child consume sugar on a daily basis, including sweets, jams, marmalades, soft drinks, fruit juices, cakes, and other sweets (such as pastries, honey, and sweetened or flavoured yoghurts, among others)?” The answers ranged from “Rarely or never” to “Once a day”, “Twice a day”, and “Three or more times a day”.

Regarding breastfeeding, we asked whether breastfeeding was exclusive, mixed, and up to what age it was carried out in months and the number of night feedings at 12 and 18 months (number: 2 times, 3 times). The Spanish version of the Early Childhood Oral Health Impact Scale (ECOHIS) was used [17,18]. The ECOHIS is a rough measure that considers parents or caregivers as key in the treatment, decision making, and perception of health conditions. It consists of 13 questions designed to evaluate the impact of problems related to oral treatment experiences on OHRQoL of preschoolers aged 2–5 years and their families. The ECOHIS questionnaire is divided into two subscales: child impact section (CI) and family impact section (FI) that were evaluated by means of two different subscales answered by the parents. The questions are answered on a Likert scale in which 1 = never, 2 = almost never, 3 = occasionally, 4 = frequently, and 5 = very frequently. A high score on the ECOHIS scale suggests an unfavourable OHRQoL. Cronbach’s alpha for this scale was 0.70.

### 2.3. Statistical Analysis

We performed descriptive analyses, Pearson correlations, and moderation models. Subsequently, a PROCESS module (version 3.3) by Hayes was used to perform multiple simple moderation analyses (model 1) using Statistical Package for the Social Sciences (SPSS) to define the effect of the number of breastfeeding sessions at 18 months on the relationship between the DMFT index and the child’s OHRQoL.

## 3. Results

### 3.1. Sociodemographic Variables of the Infant and Mother

We assessed 273 children and their mothers. A total of 56.70% of the children were girls, and 4% were boys. They ranged in age from 2 to 4 years (M = 3.01; SD = 0.82). Specifically, 33.00% were 2 years old, 31.40% were 3 years old, and 33.50% were 4 years old. The infants generally lived with both parents (79.90%), and the rest exclusively lived with the mother (12.90%), the father (6.20%), or the grandparents (1.00%).

In reference to the descriptive statistics of the mothers, they were between 17 and 44 years old (M = 32.86; SD = 6.21). Of the mothers, 64.90% had paid employment (see Table 1). Fifty-seven percent had practised cosleeping for 3 to 36 months (M = 14.13; SD = 9.26). Mothers generally breastfed exclusively (59.30%) or performed mixed feeding (40.70%).

As can be seen in Table 1, of the mothers, most of them breastfed at night, and very few of them brushed their children’s teeth after night-time feedings.

Regarding dental hygiene, most of the mothers used gauze or a thimble and generally did so before the age of 6 months. Most mothers brushed their children from one year of age, once a day, using fluoride paste. Most of the children had visited the dentist for the first time when they were one year old. As for sugar consumption, it was observed that the majority consumed sugar once a day.

### 3.2. Relations between the Variables Studied

The relationships between the variables (duration of cosleeping, number of breastfeeding sessions at 12 months, number of breastfeeding sessions at 18 months, impact on the infant’s OHRQoL, DMFT index, age at first visit, and frequency of visits) were analysed. It was observed that a longer practice time for cosleeping was associated with an increase in breastfeeding sessions (at 12 and 18 months), a higher impact on the quality of life of the child, a higher family impact, and a higher DMFT index (Table 2). Additionally, a higher family impact was associated with a later visit to the dentist (rx = 0.15, *p* = 0.044). Similarly, a higher DMFT index was associated with a higher impact on the children’s OHRQoL and a later age of dental visits (rx = 0.22, *p* = 0.002) (Table 2).

### 3.3. Moderation Model

We evaluated the moderating effect of the number of breastfeeding sessions at 18 months (Table 3). The results indicated that the moderating effect explained 41% of OHRQoL in children, and the interaction between the number of breastfeeding sessions at 18 months and the DMFT index significantly increased the coefficient of determination (F = 44.81; ΔR^2^ = 0.41; *p* = 0.0001) (Table 3). In terms of conditional effects, the effect of DMTF index on OHRQoL was significant when one (t = 7.91; *p* < 0.0001; 95% CI = (0.92, 1.52)), two (t = 10.07; *p* < 0.0001; 95% CI = (0.67, 1.00)) and three breastfeeding sessions were performed (t = 5.66; *p* < 0.0001; 95% CI = (0.30, 0.62)).

Thus, the relationship between the DMFT index and OHRQoL is only moderated by night-time breastfeeding when the number of night-time sessions is equal to or higher than three.

When the DMFT index is 0 or 1, children who have one breastfeeding session have a better OHRQoL (oral quality of life) than those who have two or three sessions.

## 4. Discussion

Despite the fact that our sample was enrolled from a preventive programme carried out to implement oral hygiene practices in infants and that the participants, therefore, received early dental care, the recommendations of the American Academy of Pediatric Dentistry (AAPD) indicate that all children should visit the dentist for the first time before the age of 12 months [19]. In our study, 31.9% of the subjects had never been to the dentist or had visited the dentist after one year. This situation reaffirms the need for advice from health care providers who are most in touch with families at these early ages, such as nurses and paediatricians.

It is well known that cosleeping is a means to facilitate the longer duration and performance of breastfeeding [20]. This evidence is consistent with our findings, which clarify that infants who are cosleeping with their mothers have more night-time feedings but have poorer oral hygiene. In other words, the prevalence of tooth decay appears to be higher among breastfed children who feed for longer, probably because the flow of saliva is reduced at night and therefore, their buffering capacity is also reduced, which in turn increases the acidity of the mouth and, as a result of poor oral hygiene, increases the risk of tooth decay.

The OHRQoL in children has been associated with childhood caries in previous literature [21]. However, until our study, it had not been established what effects prolonged night-time breastfeeding and cosleeping have on OHRQoL. In our research, the impact on OHRQoL is relevant only for infants who have two or more night-time breastfeeding sessions, starting at 18 months of age. Night-time breastfeeding had already been described as a risk behaviour in previous studies [22]. However, until our study, under what conditions night-time breastfeeding impacts the OHRQoL of the child and their families had not been established. It is important to evaluate this variable in children since, as pointed out in [23], a low OHRQoL in the child can cause oral problems (pain in the mouth or discomfort when eating hot or cold foods), functional limitations (inability to chew or pronounce certain words or impede daily activity or schooling), psychological problems (derived from insomnia or frustration), and problems with the child’s self-image or social interaction (such as stopping smiling or talking).

The average score of the children’s OHRQoL, assessed in our sample with the ECOHIS instrument, was similar to the results obtained in other studies such as the recent work of Contaldo and collaborators (2020) [24] carried out in Italy, and others carried out in Hong Kong [25] or India [26]. This ECOHIS instrument has been widely used to measure OHRQoL in young children, showing validity and reliability [24,25,26,27].

In our study, the main problems in children with the greatest impact on OHRQoL were related to oral symptoms, as described in other works with samples of subjects of similar ages [3,28]. In relation to the impact on the family, the main effects are the feeling of guilt and the loss of family time as a result of attending to the infant’s oral problems [29].

### Limitations

First, although there is previous work that has identified risk factors that affect the development of dental caries in preschool children, which involve a complex interaction of biological, social, and economic factors, we, for obvious reasons of limitation, have demarcated the elements that are most appropriate for this research. Even so, the results are compatible with previous studies carried out in Spain in a population of the same age range [30]. Second, we used a convenience sample, which came from a specific segment of the child population of the Community of Madrid, which must be taken into account when extrapolating the results.

A possible third limitation comes from the use of self-report measures, which can be affected by memory bias and responses based on social desirability. It is possible that mothers’ recall of their past experiences with breastfeeding, hygiene, and diet may be incomplete or inaccurate. In addition, the parents answered the questionnaire with regard to the child’s OHRQoL since, although the child’s opinion is the most valuable, due to his or her age, there are certain factors that can compromise the reliability and validity of a child’s responses to the OHRQoL questionnaire. Some of these factors include short-term memory, a strong influence of recent incidents, lack of a fully developed long-term perspective, language problems during interviews, and reading problems when completing a written questionnaire. Fourth, the diagnosis of caries without interproximal radiographs may also be biased by a false negative for caries in some cases, although many of the infants had diastemata, partly justifying the non-use of this test. Finally, since the mothers reported being very sleepy or in a semiconscious state during night feedings, we cannot rule out the possibility that some of the night-time breastfeeding sessions were not recorded.

This study provides important applications for dental practices. Information on the factors that contribute to the aetiology of dental caries and its impact on OHRQoL provides a valuable tool for the planning and implementation of oral-health-promotion programmes. From pregnancy onwards, the dentist should explain to parents the benefits of breastfeeding and cosleeping but clearly explain the importance of oral hygiene for the baby (the need to remove milk residue with a silicone thimble or wet gauze after feedings to prevent the proliferation of bacteria and fungi should be explained; also, with the eruption of the first tooth, an age-appropriate brush with fluoride paste (1000 ppm in small amount should be used)) and the avoidance of added and free sugars in the feeding of infants and young children.

In short, we must convey all the benefits of breastfeeding and emphasise oral prevention from the first months of life and raise awareness and involve families, schools, and health services in this process since the consequences of poor oral health among preschool children go beyond the dental problems themselves and can also cause aesthetic, psychological, and social damage.

## 5. Conclusions

A longer practice time for cosleeping was associated with an increase in breastfeeding sessions, a higher impact on OHRQoL, higher family impact, and a higher DMFT index. More than three night-time breastfeeding sessions moderate the relationship between the DMFT index and the child’s OHRQoL. Future lines of research on the prevalence of dental caries and their impact on OHRQoL are fundamental to identifying risk groups and carrying out specific interventions to promote oral health. In addition, the measurement of OHRQoL provides relevant practical information for health authorities in charge of public health policies for the control of factors and interventions that may affect children’s oral health.

## Figures and Tables

**Table 1 children-08-00969-t001:** Descriptive statistics of sociodemographic variables of the participants’ mothers.

Variable	Category	%			%
Educational Level	Without studies	4.6			
	Primary	19.6			
	Secondary	43.3			
	Higher	32.5			
Socioeconomic Level	Low	6.7			
	Low–Medium	9.3			
	Medium	63.9			
	Medium–High	12.4			
	High	7.7			
Night-time breastfeeding	Yes	60.3	Tooth brushing after night-time feeding	Yes No	7.7 92.3
	No	39.7			
Use gauze or thimble	Yes	65.5	Start of the use of gauze or thimble	6–9 months 12 months or more	74.6 25.4
	No	34.5			
Start of tooth brushing	6 months	19.6			
	1 year 2 years 3 years	52.1 25.8 2.6			
Frequency of brushing	Once a day	53.6		53.6	53.6
	Twice a day	25.3		25.3	25.3
	Three times a day	21.7		21.7	21.7
Use fluoride paste	Yes	73.7		73.7	73.7
	No	26.3			
First visit to the dentist	6 months	23.2		53.6	53.6
	1 year	44.8		25.3	25.3
	2 years	21.1		21.7	21.7
	3 years or more	9.3			
Daily sugar intake	Rarely	29.9		53.6	53.6
	Once a day	35.1		25.3	25.3
	Twice a day	32		21.7	21.7
	Three or more times a day	3			

**Table 2 children-08-00969-t002:** Associations between variables under study.

	DC	DMFT Index	N12	N18	CI	IF
DC	1					
DMFT index	0.504 **	1				
N12	0.406 **	0.492 **	1			
N18	0.411 **	0.473 **	0.411 **	1		
OHRQoL	0.360 **	0.600 **	0.396 **	0.108	1	
IF	0.310 **	0.453 **	0.210 **	0.473 **	0.648 **	1

Note: DC = Duration of cosleeping; N12 = number of breastfeeding sessions at 12 months; N18 = number of breastfeeding sessions at 18 months; OHRQoL = children’s oral-health-related quality of life; IF = impact on family. **. Significant at the 0.01 level.

**Table 3 children-08-00969-t003:** Moderating effects of number of breastfeeding sessions at 18 months on the relationship between DMFT and impact on quality of oral life.

	Effect	SE	t	*p*	LLCI	ULCI
Model R^2^ = 0.41; F = 48.81; *p* ≤ 0.00001						
Number of breastfeeding sessions at 18 months	0.65	0.27	2.41	≤0.05	0.12	1.17
DMFT index	1.59	0.23	6.67	≤0.00001	1.12	2.06
Number of breastfeeding sessions at 18 months * DMFT index	−0.38	0.09	4.18	≤0.00001	−0.56	−0.20
Conditional effects						
One session	1.21	0.15	7.91	≤0.0001	0.91	1.52
Two sessions	0.84	0.08	10.07	≤0.0001	0.67	1.00
Three sessions	0.46	0.08	5.66	≤0.0001	0.30	0.62

Note. Bootstrap samples = 10,000. *R*^2^ = coefficient of determination. SE = standard error. LLCI = lower level of the 95% confidence interval. ULCI = upper level of the 95% confidence interval.

## Data Availability

The data that support the findings of this study are available on request from the corresponding author. The data are not publicly available due to privacy and ethical restrictions.

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
