# Peer review of "Oral-Health-Related Quality of Life and Cosleeping: The Role of Nocturnal Breastfeeding"

_children, 2021, doi:10.3390/children8110969_

Round 1

Reviewer 1 Report

The paper entitled "ORAL HEALTH-RELATED QUALITY OF LIFE AND COSLEEPING. ROLE OF NOCTURNAL BREASTFEEDING" is an original paper aiming to analyze the association between co-sleeping and the number of breast-11 feeding sessions in infants, the impact OHRQoL of the child and the impact family, and the DMFT 12 Index of the child.

The ABSTRACT is well written, clear and complete. 

The INTRODUCTION is adequate in length and content and well define the scenario and the aim of the study. 

In M&M, authors first reported the age range of enrolled subjects was from 3 to 5 years old (line 90), but then, in the inclusion criteria (Line 104), they wrote: "from 2-4 years". Please, uniform the inclusion criteria. 

The RESULTS are well and presented. In Table 1, please specify in the Table title if data are related to both parents or only mothers (I think "mothers" but needs to be clarified for the reader).

Another point to better define is the concept of "sugar consumption": did the authors consider any differences in the source of sugar, or did they group any source of any type of sugar in a single voice?

About "Number of breastfeeding": is there a range, a cut-off of numbers of night feeds per night/day over or under to define the differences reported or did the authors consider dichotomic yeas/not night feed along a period of the age of the child? (Maybe the answer is partially in the lines 118-122 when authors wrote:"With regard to breastfeeding, we asked whether it was carried out exclusively, mixed and, in the case of exclusive breastfeeding, up to what age it was done in months and the number of night feeds at 12 and 18 months.", and in line 183-186, about the night sessions of breastfeeding). Please, clarify it, at least in M&M or the first part of the results. 

I appreciated the discussion and the "limitations" part: well written, with explicit and meaningful content, with the right amount of criticism for the work done. 

In discussion, the authors could also consider (e.g. in part between lines 227-232) the following paper, about the validation and value of the ECOHIS, reporting one of the most recent works about it:

Contaldo M, Della Vella F, Raimondo E, Minervini G, Buljubasic M, Ogodescu A, Sinescu C, Serpico R. Early Childhood Oral Health Impact Scale (ECOHIS): Literature review and Italian validation. Int J Dent Hyg. 2020 Nov;18(4):396-402. DOI: 10.1111/idh.12451. Epub 2020 Jul 12. PMID: 32594620

Conclusions: 

As a personal opinion, I retain the last sentence of "limitation" (lines 277-281) could also fit at the end of the conclusion. It is not mandatory, but for a final sentence of the conclusions, the authors could consider reporting again and, in other words, the need for more attention to early childhood oral health and the need for more appropriate communications among paediatricians, pediatric dentists and families. 

Moderate English changes are required. Authors should benefit form Grammarly or other free tools to correct the hard syntax in some points 

Author Response

# Reviewer 1: The paper entitled "ORAL HEALTH-RELATED QUALITY OF LIFE AND COSLEEPING. ROLE OF NOCTURNAL BREASTFEEDING" is an original paper aiming to analyze the association between co-sleeping and the number of breast-11 feeding sessions in infants, the impact OHRQoL of the child and the impact family, and the DMFT 12 Index of the child.

The ABSTRACT is well written, clear and complete.

The INTRODUCTION is adequate in length and content and well define the scenario and the aim of the study.

In M&M, authors first reported the age range of enrolled subjects was from 3 to 5 years old (line 90), but then, in the inclusion criteria (Line 104), they wrote: "from 2-4 years". Please, uniform the inclusion criteria.

It has been reviewed and modified in the “Materials and Methods” section.

The RESULTS are well and presented. In Table 1, please specify in the Table title if data are related to both parents or only mothers (I think "mothers" but needs to be clarified for the reader).

Thank you for your appreciation. This information has been added to table 1.

Another point to better define is the concept of "sugar consumption": did the authors consider any differences in the source of sugar, or did they group any source of any type of sugar in a single voice?

This issue has been clarified in the “Materials and Methods” section, following the reviewer's indications

With regard to diet, data on added and free sugar daily intake (from rarely or never to 3 times or more per day) were collected.  Does your child consume sugar on a daily basis, including sweets, jams, marmalades, soft drinks, fruit juices, cakes, and other sweets (such as pastries, honey, and sweetened or flavored yogurts, among others)? Rarely or never, Once a day, Twice a day, Three or more times a day

About "Number of breastfeeding": is there a range, a cut-off of numbers of night feeds per night/day over or under to define the differences reported or did the authors consider dichotomic yeas/not night feed along a period of the age of the child? (Maybe the answer is partially in the lines 118-122 when authors wrote: "With regard to breastfeeding, we asked whether it was carried out exclusively, mixed and, in the case of exclusive breastfeeding, up to what age it was done in months and the number of night feeds at 12 and 18 months.", and in line 183-186, about the night sessions of breastfeeding). Please, clarify it, at least in M&M or the first part of the results.

About the number of feedings, we asked them:

How many times was your child breastfeeding at night at 12 months?

How many times was your child breastfeeding at night at 18 months?

The mother answered with a number (2 times, 3 times...etc).

I appreciated the discussion and the "limitations" part: well written, with explicit and meaningful content, with the right amount of criticism for the work done.

In discussion, the authors could also consider (e.g. in part between lines 227-232) the following paper, about the validation and value of the ECOHIS, reporting one of the most recent works about it:

Contaldo M, Della Vella F, Raimondo E, Minervini G, Buljubasic M, Ogodescu A, Sinescu C, Serpico R. Early Childhood Oral Health Impact Scale (ECOHIS): Literature review and Italian validation. Int J Dent Hyg. 2020 Nov;18(4):396-402. DOI: 10.1111/idh.12451. Epub 2020 Jul 12. PMID: 32594620.

We have read and included the study suggested by the reviewer. Thank you for the suggestion, it helps us to keep the manuscript up to date.

Conclusions:

As a personal opinion, I retain the last sentence of "limitation" (lines 277-281) could also fit at the end of the conclusion. It is not mandatory, but for a final sentence of the conclusions, the authors could consider reporting again and, in other words, the need for more attention to early childhood oral health and the need for more appropriate communications among paediatricians, pediatric dentists and families.

We have changed this sentence at the end of the conclusions. Thank you.

Moderate English changes are required. Authors should benefit from Grammarly or other free tools to correct the hard syntax in some points

Changes have been made to the manuscript to make it easier to understand. A native researcher has rechecked the manuscript. Thank you.

Reviewer 2 Report

The present observational study, concerning the association between co-sleeping, number of breast- feeding sessions, the impact OHRQoL of the child and the impact family, and the DMFT of the child may contribute to the aetiology of dental caries in children, and provide important applications for dental practice and for oral health promotion programs in childhood.

Manuscript structure is well organized.

Editing for English language is needed.

My main concerns are reported below:

INTRODUCTION:

  • Please re-prase the sentence in lines 75-76 "they may decide to share the bed with the child once they have acquired strategies to do so with minimal risk to the infant's oral health".

MATERIALS AND METHODS:

  • Please re-prase the periods delucidating inclusion and exclusion criteria in lines 101-105
  • Bedding or Bed-sharing? (lines 118 and 119)
  • Please re-prase the sentence in lines 134-135
  • Please, explain the acronym "SPSS" (line 137).

RESULTS:

  • Please, add informations regarding boys (lines 141-142)
  • Please, add to Table 1 descriptive information written in the text and re-phrase periods in lines 146-147
  • Please, re-phrase the sentence in lines 190-191.

Author Response

SPECIFIC COMMENTS: 

# REVIEWER 2:

The present observational study, concerning the association between co-sleeping, number of breast- feeding sessions, the impact OHRQoL of the child and the impact family, and the DMFT of the child may contribute to the aetiology of dental caries in children, and provide important applications for dental practice and for oral health promotion programs in childhood.

Manuscript structure is well organized.

Editing for English language is needed.

My main concerns are reported below:

INTRODUCTION:

Please re-prase the sentence in lines 75-76 "they may decide to share the bed with the child once they have acquired strategies to do so with minimal risk to the infant's oral health".

We have changed this sentence.

MATERIALS AND METHODS:

Please re-prase the periods delucidating inclusion and exclusion criteria in lines 101-105

The information has been added in the material and method section.

Bedding or Bed-sharing? (lines 118 and 119)

We have changed this concept to bed-sharing.

Please re-prase the sentence in lines 134-135

We have changed this sentence.

Please, explain the acronym "SPSS" (line 137).

We have explained the acronym.

RESULTS:

Please, add informations regarding boys (lines 141-142)

Sorry, this was a translation error, the data presented is general. We have modified the paragraph to make it more understandable.

Thank you for your feedback.

Please, add to Table 1 descriptive information written in the text and re-phrase periods in lines 146-147

The descriptive information has been included in table 1 and in the text. We have re-phrased periods in line 146-147 following the reviewer's indications

Please, re-phrase the sentence in lines 190-191.

We have changed this sentence.